# Impact of a team-based versus individual clinician-focused training approach on primary healthcare professionals' intention to have serious illness conversations with patients: A theory-informed process evaluation embedded within a cluster randomized trial

Lucas Gomes Souza[1,2], Patrick M. Archambault[2,3,4], Dalil Asmaou Bouba[1,2], Suélène Georgina Dofara[2], Sabrina Guay-Bélanger[2], Sergio Cortez Ghio[2], Souleymane Gadio[2], Shigeko (Seiko) Izumi[5], LeAnn Michaels[6], Jean-Sébastien Paquette[2,4,7], Annette M. Totten[6], France Légaré[2,4,8]*, The Meta-LARC ACP Cluster Randomized Trial team[¶]

1 Department of Social and Preventive Medicine, Faculty of Medicine, Université Laval, Québec, Quebec, Canada, 2 VITAM – Centre de recherche en santé durable, Centre intégré universitaire de santé et de services sociaux de la Capitale-Nationale, Québec, Quebec, Canada, 3 Centre de recherche intégrée pour un système apprenant en santé et services sociaux, Centre intégré de santé et de services sociaux de Chaudière-Appalaches, Lévis, Quebec, Canada, 4 Department of Family Medicine and Emergency Medicine, Faculty of Medicine, Université Laval, Québec, Quebec, Canada, 5 School of Nursing, Oregon Health & Science University, Portland, Oregon, United States of America, 6 Department of Medical Informatics & Clinical Epidemiology, School of Medicine, Oregon Health & Science University, Portland, Oregon, United States of America, 7 ARIMED Laboratory, GMF-U du Nord de Lanaudière, CISSS Lanaudière, Lanaudière, Quebec, Canada, 8 Centre de recherche du Centre hospitalier universitaire de Québec, Québec, Quebec, Canada

¶ Membership of The Meta-LARC ACP Cluster Randomized Trial team is provided in the acknowledgements.
* France.Legare@fmed.ulaval.ca

## Abstract

### Background

Cluster randomized trials (cRTs) on the effectiveness of training programs face complex challenges when conducted in real-world settings. Process evaluations embedded within cRTs can help explain their results by exploring possible causal mechanisms impacting training effectiveness.

### Objective

To conduct a process evaluation embedded within a cRT by comparing the impact of team-based vs. individual clinician-focused SICP training on primary healthcare professionals' (PHCPs) intention to have serious illness conversations with patients.

**Data availability statement:** All relevant data are within the manuscript and its Supporting information files.

**Funding:** Research reported in this paper was funded through a Patient-Centered Outcomes Research Institute® (PCORI®) Award (PLC-1609-36277) The results presented in this paper are solely the responsibility of the authors and do not necessarily represent the views of the Patient-Centered Outcomes Research Institute® (PCORI®), its Board of Governors or Methodology Committee. FL holds the Tier 1 Canada Research Chair in Shared Decision Making and Knowledge Mobilization (Funder Grant: N/A). LGS received the scolarship Fonds stratégique de développement de la recherche from VITAM-centre de recherche en santé durable (Funder Grant: N/A). PA received a Fonds de recherche du Québec – Santé Clinical Scholar Award.The funders did not play any role in the study design, data collection and analysis, decision to publish, or preparation of the manuscript.

**Competing interests:** The authors have declared that no competing interests exist.

## Methods

The cRT involved 45 primary care practices randomized into a team-based (intervention) or individual clinician-focused (comparator) training program and measured primary outcomes at the patient level: days at home and goals of care. To perform this theory-informed mixed-methods process evaluation embedded within the cRT, a different outcome was measured at the level of the PHCPs, namely, PHCPs' intention to have serious illness conversations with patients as measured with CPD-Reaction. Barriers and facilitators to implementing the conversations were identified through open-ended questions and analyzed using the Theoretical Domains Framework. The COM-B framework was used to triangulate data. Results were reported using the CONSORT and GRAMMS reporting guidelines.

## Results

Of 535 PHCPs from 45 practices, 373 (69.7%) fully completed CPD-Reaction (30.8% between 25-34 years old; 78.0% women; 54.2% had a doctoral degree; 50.1% were primary care physicians). Mean intention scores for the team-based (n=223) and individual clinician-focused arms (n=150) were 5.97 (standard error (SE): 0.11) and 6.42 (SE: 0.13), respectively. Mean difference between arms was 0.0 (95% CI -0.29; 0.30; p=0.99) after adjusting for age, education and profession. The team-based arm reported barriers with communication, workflow, and more discomfort in having serious illness conversations with patients.

## Conclusions

Team-based training did not outperform individual clinician-focused in influencing PHCPs' intention to have serious illness conversations. This process evaluation suggests that team-based training could improve intervention effectiveness by focusing on interprofessional communication, better organized workflows, and better support and training for non-clinician team members.

**Registration**: ClinicalTrials.gov (ID: NCT03577002).

## Introduction

Cluster Randomized Trials (cRTs) using a pragmatic approach aim to evaluate the effectiveness of interventions in real-world clinical settings, providing advantages such as enhanced generalizability and relevance to routine care [1]. However, pragmatic cRTs present unique challenges. Implementing and maintaining interventions in cRTs within the complex environment of routine clinical care is difficult due to variations in clinical practice, healthcare systems, and participant populations [2]. Process evaluations embedded within cRTs are therefore increasingly used to shed light on the 'black box' of complex interventions and to provide information that helps interpret outcome results and move them into practice [3,4].

In 2017, the Meta-Network Learning and Research Center Advanced Care Planning (Meta-LARC ACP) cRT team designed a comparative effectiveness cRT to assess two training approaches for the Serious Illness Care Program (SICP) developed by Ariadne Labs [5,6]. The cRT compared a team-based training approach to conducting the SICP (intervention) with the traditional clinician-based training approach (comparator) [7]. The rationale for this cRT

stemmed from the initial assumption in the design of the SICP, that only individual clinicians such as physicians, nurse practitioners, or physician assistants, are responsible for having serious illness conversations with patients. However, insights from studies on team approaches to chronic and complex care suggest that adopting a team-based approach could facilitate the implementation of serious illness conversations within clinical practice [8,9].

In recognition of the pivotal role played by primary care teams in patient care, the cRT explored a training approach that integrated task-sharing strategies. The shared tasks included patient identification, conversation preparation, discussion initiation, and follow-up [8,10]. The team approach aligned with systematic reviews which identified time constraints as a significant barrier to implementing these discussions in primary care settings. The expectation was that involvement of multiple primary healthcare professionals (PHCPs) could facilitate implementation by reducing the time commitment required from each individual [11,12]. However, evidence supporting the efficacy of the team-based approach in primary care was limited at the time the cRT was conceived [13]. This underscored the need to evaluate the approach in a real-world primary care setting.

Process evaluations embedded within cRTs aim to provide insights into how an intervention was delivered, how participants received it, and whether it was implemented as intended [4]. In the parent Meta-LARC ACP cRT, the impact of a team-based training approach compared to a clinician-based training approach was measured using patient centered outcomes [7]. In this process evaluation, measuring the impact of the intervention on PHCPs themselves instead of patients and, specifically, on PHCPs' intention to have serious illness conversations with patients was necessary to provide insights into how the team-based training approach was delivered, received and implemented in the parent cRT.

A process evaluation focusing on PHCPs' behavioral intention, and its psychosocial determinants would thus provide information about key factors affecting effective implementation of the intervention and could inform recommendations for its improvement. Key factors would include the modifiable psychosocial factors that influence PHCPs' intention to have serious illness conversations, and barriers and facilitators that they perceive to implementing the conversations. Thus, to influence behavior through behavioral intention, one would need to modify psychosocial factors using known behavior change techniques [14]. No studies have specifically addressed the factors that drive PHCPs' intentions to have serious illness conversations with patients. Understanding these components is crucial for designing training interventions that effectively promote the adoption of this important behavior in clinical practice.

According to several socio-cognitive theories, behavioral intention is the central factor influencing the adoption of a given behavior and is therefore an acceptable surrogate for behavior change [15–19]. It provides a more immediate and cost-effective measure of actual clinical behavior change and is widely used to evaluate the efficacy of professional development training interventions [20]. Measuring intention would thus allow for a targeted assessment of the intervention's influence on PHCPs, the primary implementers of the intervention in the parent cRT. It could also provide information on intervention strengths and weaknesses to inform the interpretation and adaptation of future SICP interventions [21]. Ultimately, it could contribute to a the cumulative implementation science knowledge base.

In order to achieve these aims, three established socio-cognitive theories of behavior change were incorporated into this process evaluation [14,16,22,23]. These theoretical frameworks played key roles in identifying and addressing specific implementation challenges, enhancing our understanding of the intervention, and suggesting modifications that could increase the impact of future such interventions [21].

We hypothesized, based on our previous theory-informed studies on interprofessional interventions [24,25], that a team-based training would outperform an individual

clinician-focused training in influencing PHCPs'intention to have serious illness conversations with their patients. These studies demonstrate that an interprofessional team-based model could pottentiallypromote positive intentions to have serious illness conversations with patients. Furthermore, these studies on interprofessional interventions elucidated the modifiable psychosocial factors that positively correlate with intention within an interprofessional context [24,25].

Therefore, we performed a theory-based process evaluation embedded within the parent cRT to assess the effect of the two SICP training approaches on PHCPs' intention to have serious illness conversations with patients and to identify potential facilitators and barriers to the adoption of this behavior.

## Materials and methods

### Study design and setting

This study is a mixed-methods concurrent embedded process evaluation embedded within a cRT [26–28]. Secondary post-intervention qualitative and quantitative data from the parent cRT were used in the analysis. Process evaluations often employ mixed methods designs which provide a more comprehensive and multifaceted understanding of intervention implementation [21,29]. We used the integrative model of behavior prediction in healthcare professionals by Godin et al. for our quantitative analysis [15,16,30], the Theoretical Domains Framework (TDF) for our qualitative analysis [22,31], and the Capability, Opportunity, Motivation and Behavior (COM-B) model to triangulate findings [32]. We reported this study following the extension of Consolidated Standards of Reporting Trials (CONSORT) for cRTs (S1 Checklist) and followed the Good Reporting of a Mixed Methods Study (GRAMMS) checklist (S2 Checklist) [33,34]. The parent cRT (subject of the process evaluation) is registered at ClinicalTrials.gov (NCT03577002). The protocol is published elsewhere [7] and the results for the cRT primary outcomes at the patient level (days spent at home and goal concordant care) are under review.

The parent Meta-LARC ACP cRT was conducted in community-based primary care practices in five US states (Colorado, Iowa, North Carolina, Oregon, Wisconsin) and two Canadian provinces (Quebec and Ontario) recruited through Meta-LARC, a consortium of practice-based research networks (PBRNs) [7]. A cRT design was selected as the knowledge learned in the SICP trainings was to be implemented at a practice level.

The Medical Research Council framework [29] was used to guide the process evaluation. This framework suggests that the context and mechanisms of impact of an intervention are the key functions of a process evaluation. To explore the main mechanisms of impact, this study measured PHCPs' intention (and its modifiable psychosocial factors) to adopt the target behavior, defined as *having serious illness conversations with patients*. A previous study focused on the development process of the intervention [35]. Another study explored the contextual factors that influenced the sustainability of its impacts, particularly in the context of the COVID-19 pandemic [36]. This combined approach allows to address all the different aspects of a process evaluation [29].

### Ethical approval

The parent Meta-LARC ACP cRT was approved by the Trial Innovation Network Single Institutional Review Board at Vanderbilt University Medical Center (#181084) for the U.S. sites; by the Research Ethics Board of the Centre Intégré Universitaire de Santé et de services sociaux (CIUSSS) de la Capitale-Nationale in Quebec City, Canada (#MP-13-2019-1526) for the sites in Quebec, and by the Health Sciences Research Ethics Board of the University of Toronto

(#36631) for the sites in Ontario. Data and outcomes for this process evaluation were included in the approval. All subjects willingly agreed to take part in the study, and their consent (verbal or written) was obtained and registered by PBRNs in accordance with the regulations of the Institutional Review Board or Research Ethics Board in effect.

## Participants and eligibility

To participate in the parent cRT a practice had to a) be willing and able to be randomized to either the team-based or individual-clinician-based SICP approach, b) have sufficient staff to participate in the team-based arm, and c) not be engaged in another standardized advanced care planning (ACP) program. Detailed criteria for practice eligibility are published elsewhere [7]. In this process evaluation, we analyzed a secondary outcome, the impact of the intervention on PHCPs' behavioral intention, and therefore participants for this process evaluation were the PHCPs (primary care clinicians and other primary care team members) working in the primary care practices enrolled in the parent cRT. PHCPs from participating eligible practices were invited to participate in the training and to answer the after-training questionnaires, but not required to participate.

## Randomization

The units of randomization were the primary care practices stratified by PBRN. To assure allocation concealment, involvement in randomization was limited to statisticians not involved in other aspects of the project. Staff at the PBRNs, practices, and the Meta-LARC coordinating center were not involved in the randomization. Statisticians completing the analysis were blinded to allocation until the parent cRT primary outcomes analysis was completed. Investigators, PBRN leadership, practices, and research staff were not blinded to the assignment. Practices and participating PHCPs could not be blinded to which approach they were assigned, as they needed to be actively trained and implement the intervention or comparator. More details regarding randomization are published elsewhere [7].

## Intervention and comparator arms

The SICP developed by Ariadne Labs was adapted to be used by interprofessional teams of PHCPs [5]. Training lasted three hours per arm: a 1.5-hour online module (Part A), and a 1.5-hour in-person role-play session (Part B) (Fig 1). Training materials are available at https://primarycareacp.org, including the Serious Illness Conversations Guide (SICG), a tool designed by the original developers of the SICP to facilitate communication with patients with serious illnesses [5].

Recruitment and training of PHCPs took place between 26-10-2018 and 30-11-2019. A train-the-trainer model was used. The intervention proceeded as described in the study protocol [7].

## Intervention arm

The intervention arm received SICP training session adapted to a team-based approach, whereby primary care clinicians (i.e., primary care physicians, physician assistants, nurse practitioners, and medical residents) and other team members such as medical assistants, nurses, dietitians, and social workers were invited to complete the training. The adapted team-based approach was based on an interprofessional shared decision-making model and workshop [37,38] that has been used in a variety of contexts and showed a positive impact on behavioral intention [39,25]. Training involved guidance on establishing a common

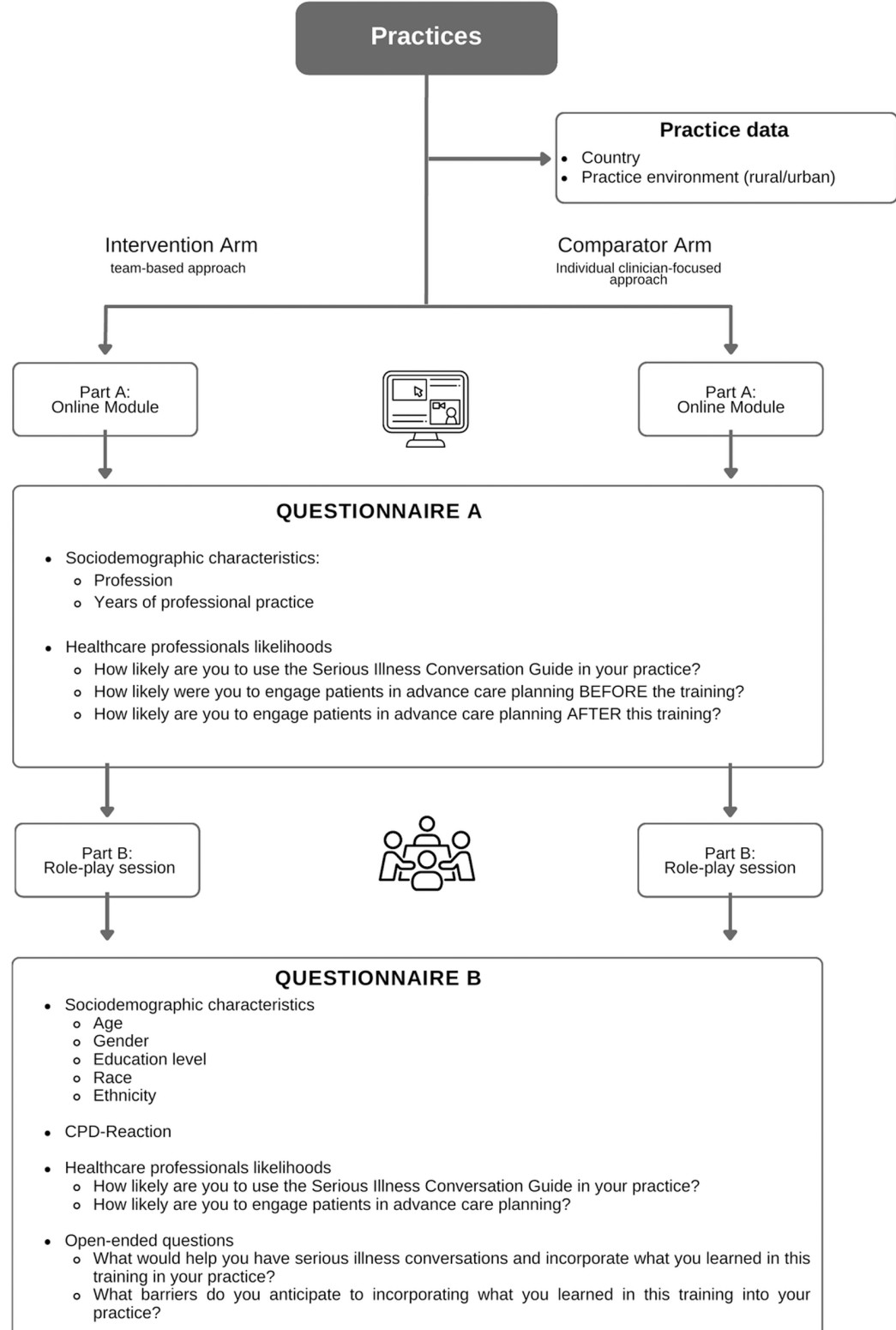

**Fig 1. Study design and data collection.**

understanding of the processes and goals of serious illness conversations, dividing and sharing tasks, recognizing each member's contributions, promoting continuous communication in the team, and recognizing the organizational or functional constraints of each profession [35].

### Comparator arm

The comparator arm received the individual clinician-focused approach as designed by Ariadne Labs [5]. It focuses only on primary care clinicians (i.e., primary care physicians, physician assistants, nurse practitioners, and medical residents). Other primary care team members, such as medical assistants, nurses, dietitians, and social workers did not participate in the training. This approach assumes that primary care clinicians alone are responsible for selecting patients, conducting, and documenting serious illness conversations.

### Outcomes and measurements

The outcome of interest for this process evaluation was the impact of our training intervention on PHCPs' intention to have serious illness conversations with patients. This focus on intention was based on its established role as a proxy for behavior and its demonstrated predictive capacity for behavior change up to six months following training interventions [15–18].

We used the CPD-Reaction tool to measure PHCPs' behavioral intention to have serious illness conversations with patients. CPD-Reaction is a self-administered, theory-informed, validated tool that assesses five psychosocial constructs (intention, beliefs about capabilities, beliefs about consequences, moral norm, and social influences) using a 12-item self-administered questionnaire [17,40]. Most items are measured using a Likert scale of 1–7, where 1 represents "strongly disagree" and 7 represents "strongly agree." One item, the social influences construct, ranges from 1 to 5 [30].

However, as proposed by Lou, Atkins and West [22], behaviors do not occur in isolation. Depth of analysis is increased by expanding the range of relevant behaviors measured while maintaining appropriate selectivity [14]. Therefore, as additional behavior-related outcomes for this process evaluation, participants were also asked about their likelihood to engage in ACP and to use the SICG. ACP was included because it is a more established, broader concept in the field that encompasses end-of-life discussions, care objectives, and their legal documentation [41]. Questions about PHCPs' likelihood to engage in ACP were presented before and after Part A, using a retrospective pretest-posttest evaluation design [42]. Questions about the likelihood to use the SICG in their practices were presented after Part B. Face and content validity of the likelihood measures were established based on studies on similar training programs with PHCPs [43,44]. These questions also used a Likert scale of 1 to 10 where 1 represented "extremely unlikely", 5 represented "moderate," and 10 "extremely likely."

Finally, the facilitators and barriers to PHCPs incorporating the knowledge acquired in their practices and to have serious illness conversations with patients were evaluated using two open-ended questions. These facilitators and barriers were mapped onto the TDF. The TDF was developed through a consensus of experts who consolidated 33 psychosocial theories of behavior change to generate 14 domains [45]. Quantitative and qualitative findings were then triangulated using the the COM-B model of behavior aiming to formulate recommendations for future training programs or SICP adaptations [14]. Fig 1. details all the process evaluation measurements and outcomes sought throughout the different steps of the intervention.

### Data collection

Data collection took place between October 2018 and November 2019. Before randomization, sociodemographic data were collected as well as additional data directly from practices

(e.g., country and practice environment) (Fig 1). This analysis uses the data collected through questionnaires administered immediately after training was completed in each practice, as we were interested in PHCPs' immediate reactions to each training approach. All questionnaires in the study were self-administered. Open-ended questions were used instead of semi-structured interviews or focus groups to explore barriers and facilitators. These decisions streamlined data collection, enhancing both efficiency and participant convenience.

## Sample size

The sample size calculation for the parent study is available in the published protocol [7]. A sample size estimate for PHCPs involved in this process evaluation study was not calculated as their responses were not the primary outcome of the parent cRT. A total of 535 PHCPs attended the trainings, 326 in the team-based and 209 in the individual clinician-focused arms. Data from 373 PHCPs (223 team-based and 150 individual clinician-focused) who attended the training and completed the intention construct in the CPD-Reaction tool were analyzed. For these 373 PHCPs, a post-hoc power of 0.75 based on an unadjusted mixed linear regression model was obtained.

## Analysis

**Quantitative analysis.** Descriptive statistics were used to report our variables including practice and participant characteristics. Categorical variables were described with absolute (n) and relative (%) frequencies; and continuous variables with their central tendency measures (mean) and dispersion (standard deviation).

Then, a mixed linear model was used to compare mean scores between study arms for PHCPs' intention to have serious illness conversations with patients. To account for clustering the practice identifier was used as a random factor when fitting the models.

Next, a bivariate mixed linear model for each sociodemographic variable of interest to examine its effect on PHCPs' intention to have serious illness conversations with patients was applied. Subsequently, a multivariable mixed linear model including all variables for which we detected a potential effect (p < 0.20) on PHCPs' intention in the bivariate analyses was used. Then manual backward stepwise selection based on variable significance for a final adjusted model was performed. The study allocation arm variable was kept in the model regardless of significance as the objective was to compare the impact of training approaches.

Finally, a mixed linear model was used to evaluate participants' perceived likelihood of engaging patients in ACP (i.e., before (reported retrospectively) and then after training), and of using the SICG in their practice after both parts of the training. The mean differences between the likelihood to engage in ACP measured before and after Part A of the training between each arm was compared, as were mean differences between the likelihood to use the SICG, also measured between each arm. The practice identifier was used as a random factor in all models. All analyses were performed with SAS (Statistical Analysis Software) 9.4.

**Qualitative analysis.** To further explore intention, using the TDF as a guide, a descriptive analysis of the answers to the two open-ended questions was performed [22,46]. Codes for thematic analysis were developed using a deductive approach by two researchers (SGD, LGS) [47]. Then, two researchers (LGS, DAB) conducted the analysis and resolved any disagreements by consensus. If the disagreement remained, a third senior researcher was consulted (FL). Data were then deductively classified into TDF domains. The frequency of each barrier and facilitator was calculated by recording the number of times it was mentioned.

**Triangulating qualitative and quantitative data.** This process evaluation aimed to propose practical theory-driven recommendations to improve the intervention. A triangulation of

qualitative and quantitative data was therefore performed for a broader understanding of the psychosocial factors influencing the targeted behavior [48]. A comparison between the five psychosocial constructs assessed in the CPD-Reaction questionnaire and the TDF domains identified was conducted. Points of convergence and divergence between quantitative and qualitative data were identified and instances where both types of data provided insights into the same constructs were explored. Recommendations were formulated utilizing the COM-B model of behavior, which posits three criteria—capacity, opportunity, motivation—for the occurrence of a behavior [14,49]. These criteria and their subcategories were linked to the TDF domains, along with their associated barriers or facilitators (S1 Fig) [50]. The COM-B model further proposes nine intervention functions, aligned with TDF domains, that promote behavior change, namely education, persuasion, incentivization, coercion, training, restriction, environmental restructuring, modeling, and enablement [14,32,51,52]. By discerning which intervention functions corresponded to the results, behavior change techniques associated with the relevant functions were identified and selected to derive recommendations [14]. The proposed recommendations were reviewed by all authors.

## Results

### Practice and PHCP characteristics and flow diagram

Table 1 describes characteristics of randomized practices and demonstrates that the randomization of clinics in the parent cRT was adequate.

Fig 2 illustrates the flow of practices and PHCPs in this process evaluation study embedded within the Meta-LARC ACP cRT. Thirty-eight practices participated (19 in each arm),

**Table 1. Participating primary care practice characteristics.**

|  | Total | Team-based arm | Individual clinician-focused arm |
|---|---|---|---|
| Number of practices | 45 | 23 | 22 |
| Country, n (%) |  |  |  |
| United States | 33 (73.3) | 17 (73.9) | 16 (72.7) |
| Canada | 12 (26.7) | 6 (26.1) | 6 (27.3) |
| Size (# of primary healthcare professionals), n (%) |  |  |  |
| Small (2-5) | 8 (17.8) | 6 (26.1) | 2 (9.0) |
| Medium (6-12) | 19 (42.2) | 9 (39.1) | 10 (45.5) |
| Large (13-85) | 18 (40.0) | 8 (34.8) | 10 (45.5) |
| Geographic setting, n (%) |  |  |  |
| Rural | 20 (44.4) | 12 (52.2) | 8 (36.4) |
| Suburban | 8 (17.8) | 3 (13.0) | 5 (22.7) |
| Urban | 17 (37.8) | 8 (34.8) | 9 (40.9) |
| Ownership, n (%) |  |  |  |
| Hospital/health system | 31 (68.9) | 13 (56.6) | 18 (81.8) |
| Physician or physician group | 11 (24.4) | 7 (30.4) | 4 (18.2) |
| Federally qualified health centre | 3 (6.7) | 3 (13.0) | – |
| Specialty, n (%) |  |  |  |
| Family medicine | 34 (75.5) | 15 (68.2) | 19 (82.6) |
| Internal medicine | 8 (17.8) | 5 (22.7) | 3 (13.0) |
| Both family and internal medicine | 3 (6.7) | 2 (9.1) | 1 (4.4) |
| Size of primary healthcare professionals, median (min to max) | 10 (3 to 46) | 8 (4 to 46) | 12 (3 to 40) |

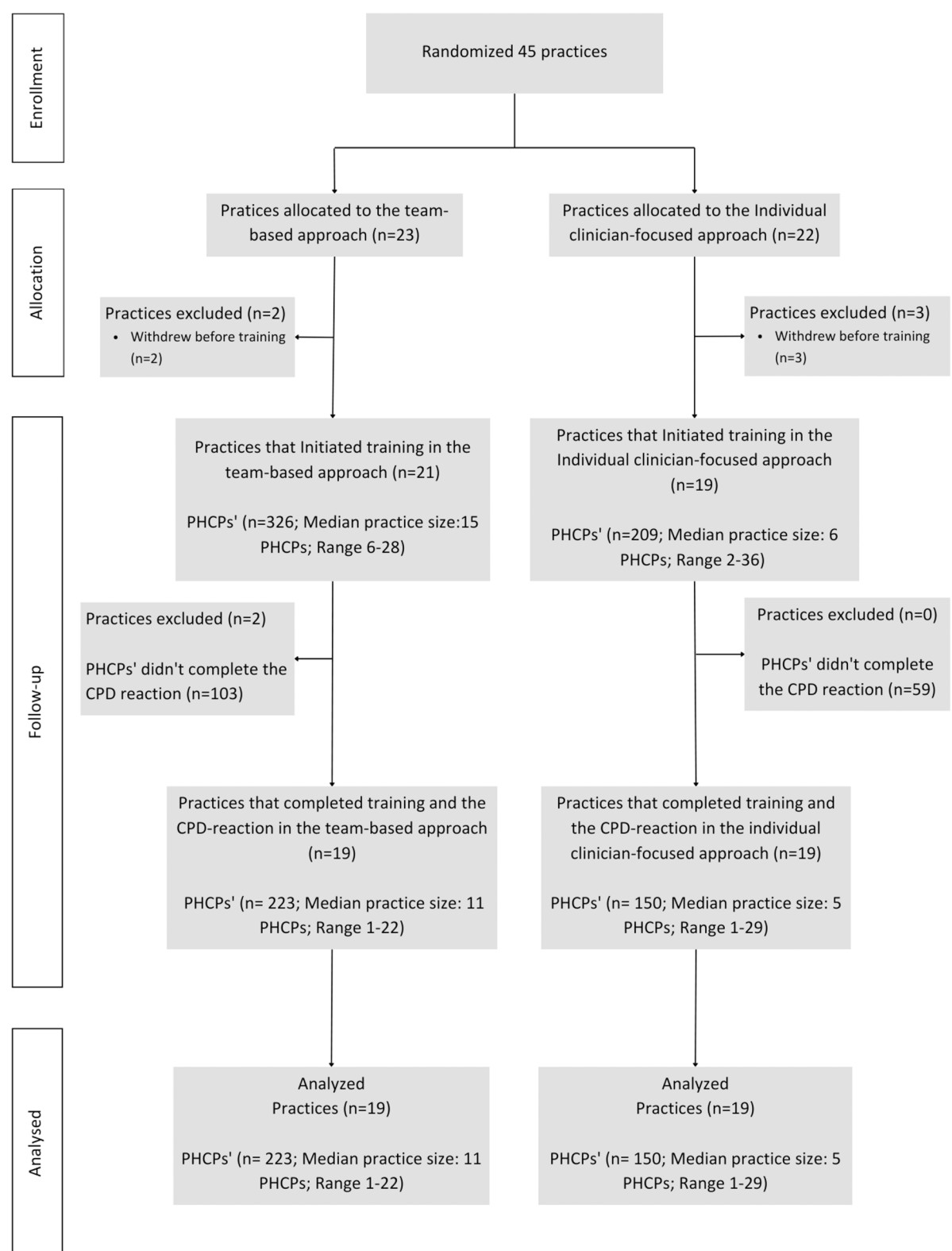

**Fig 2. Flowchart of primary healthcare professionals (PHCPs).**

**Table 2.** *Characteristics of participating PHCPs by arm.*

|  | Total | Team-based arm | Individual clinician-focused arm |
|---|---|---|---|
| Number of professionals | 373 | 223 | 150 |
| Country, N (%) | | | |
| USA | 216 (57.9) | 143 (64.1) | 73 (48.7) |
| Canada | 157(42.1) | 80 (35.9) | 77 (51.3) |
| Age, N (%) | | | |
| Missing data | 12 (3.2) | 10 (4.5) | 2 (1.3) |
| 18-24 | 11 (2.9) | 8 (3.6) | 3 (2.0) |
| 25-34 | 115 (30.8) | 64 (28.7) | 51 (34.0) |
| 35-44 | 113 (30.3) | 63 (28.2) | 50 (33.3) |
| 45-54 | 64 (17.2) | 38 (17.0) | 26 (17.3) |
| 55-64 | 42 (11.3) | 30 (13.5) | 12 (8.0) |
| > 65 | 12 (3.2) | 10 (4.5) | 2 (1.3) |
| Prefer not to answer | 4 (1.0) | – | 4 (2.7) |
| Sex, N (%) | | | |
| Missing data | 3 (0.8) | 3 (1.3) | – |
| Male | 79 (21.2) | 35 (15.7) | 44 (29.3) |
| Female | 291 (78.0) | 185 (83.0) | 106 (70.7) |
| Ethnicity, N (%) | | | |
| Missing data | 7 (1.9) | 6 (2.7) | 1 (0.7) |
| Yes, Hispanic or Latino | 13 (3.5) | 9 (4.0) | 4 (2.7) |
| No, not Hispanic or Latino | 347 (93.0) | 205 (92.0) | 142 (94.7) |
| Prefer not to answer | 6 (1.6) | 3 (1.3) | 3 (2.0) |
| Race, N (%) | | | |
| Missing data | 6 (1.6) | 5 (2.2) | 1 (0.7) |
| White | 320 (85.8) | 192 (86.1) | 128 (85.3) |
| Asian | 20 (5.4) | 13 (5.8) | 7 (4.7) |
| American Indian or Alaska Native | 2 (0.5) | 1 (0.4) | 1 (0.7) |
| Black or African American | 14 (3.7) | 9 (4.0) | 5 (3.3) |
| Hawaïan Native or Pacific Islander | 1 (0.3) | 1 (0.4) | – |
| Others | 2 (0.5) | 1 (0.4) | 1 (0.7) |
| Prefer not to answer | 8 (2.1) | 1 (0.4) | 7 (4.7) |
| Education level, N (%) | | | |
| Missing data | 4 (1.0) | 3 (1.4) | 1 (0.7) |
| Doctoral degree | 202 (54.2) | 91 (40.8) | 111 (74.0) |
| Master's degree | 65 (17.4) | 34 (15.2) | 31 (20.7) |
| Bachelor's degree | 40 (10.7) | 40 (17.9) | – |
| High school | 9 (2.4) | 9 (4.0) | – |
| Others | 53 (14.2) | 46 (20.6) | 7 (4.7) |
| Profession, N (%) | | | |
| Missing data | – | – | – |
| Primary care physician | 187 (50.1) | 86 (38.6) | 101 (67.3) |
| Nurse practitioner | 35 (9.4) | 16 (7.2) | 19 (12.7) |
| Nurse | 54 (14.5) | 54 (24.2) | – |
| Physician assistant | 16 (4.3) | 7 (3.1) | 9 (6.0) |
| Social worker | 6 (1.6) | 6 (2.7) | – |
| Medical assistant | 33 (8.9) | 33 (14.8) | – |
| Resident | 29 (7.8) | 8 (3.6) | 21 (14.0) |
| Others (Psychologists, dietitians, etc.) | 13 (3.5) | 13 (5.8) | – |

representing a participation rate of 84.4% (38/45). A total of 373 (69.7%) PHCPs fully completed the intention construct in CPD-Reaction. The sociodemographic characteristics of the participating PHCPs are detailed in Table 2.

## Quantitative results

**Behavioral intention.** Table 3 shows the scores for behavioral intention and its psychosocial determinants. On a scale of 1–7, the mean intention score was 5.97 (standard error (SE): 0.11) for the team-based arm (intervention) and 6.42 (SE: 0.13) for the individual clinician-focused arm (comparator). The difference in mean intention scores and 95% confidence interval (95% CI) between arms was -0.45 (-0.79 to -0.11), with a p-value of 0.01 (Table 4). There was no statistically significant difference in the other CPD-Reaction constructs between arms.

However, after adjusting for age, education, and profession, the difference in intention between the two groups was no longer statistically significant: 0.00 (95% CI: -0.29 to 0.30), with a p-value of 0.99 **(Table 4).**

**Likelihood to engage in ACP and to use the SICG.** The mean difference in scores between the two arms for "How likely were you to engage patients in ACP before training?" was -1.09; p = 0.01, and for "How likely are you to engage patients in ACP after training?" was -0.75 (p = 0.01) (S1 Table). Mean difference between pre- and post-training likelihood to engage in ACP was 2.95 (p < 0.001) in the team-based arm and 2.60 (p < 0.001) in the individual clinician-focused arm (Table 5), the difference between arms was not statistically

**Table 3. Comparison of intention scores and and its psychosocial determinants measured with the CPD-Reaction questionnaire after the training.**

| | | Team-based arm | | Individual clinician-focused arm | Estimate difference (95% CI)[b] | p-Value |
|---|---|---|---|---|---|---|
| Intention and psychosocial determinants– Score Range (1 to 7)[a] | N | Mean (SE) | N | Mean (SE) | | |
| **Beliefs about capabilities** | 223 | 5.40 (0.08) | 150 | 5.60 (0.09) | -0.20 (-0.45; 0.05) | 0.11 |
| **Social Influences** | 217 | 4.69 (0.13) | 147 | 4.83 (0.15) | -0.13 (-0.53; 0.26) | 0.50 |
| **Beliefs about consequences** | 223 | 6.36 (0.07) | 150 | 6.56 (0.09) | -0.21 (-0.43; 0.02) | 0.08 |
| **Moral Norm** | 223 | 6.71 (0.04) | 150 | 6.81 (0.05) | -0.10 (-0.23; 0.02) | 0.10 |
| **Intention** | 223 | 5.97 (0.11) | 150 | 6.42 (0.13) | -0.45 (-0,79; -0.11) | 0.01 |

Analyzed using a linear mixed model.

p-value < 0.05 – statistically significant (t-test);

95% CI, confidence interval at 95%; SE, standard error of the mean.

[a]The social influences construct is the only construct that has one item with a Likert scale that ranges from 1 to 5.

[b]Least squares mean

**Table 4. Comparison of the intention to have serious illness conversations with patients between the team-based arm and the individual clinician-focused arm.**

| | Unadjusted Model | | Adjusted model[b] | |
|---|---|---|---|---|
| | Mean difference (95% CI)[c] | p-Value[a] | Mean difference (95% CI)[c] | p-Value[a] |
| **Individual clinician-focused arm** | Reference | | Reference | |
| **Team-based arm** | -0.45 (-0.79; -0.11) | 0.01 | 0.00 (-0.29; 0.30) | 0.99 |

Analyzed using a linear mixed model.

[a]p-value < 0.05 – statistically significant (t-test).

[b]Adjusted for age, profession and education level,

[c]95% CI, confidence interval at 95%.

**Table 5.** *Differences in the likelihood questions about engaging patients in ACP before and after training and using the SICG.*

| | Mean (SE) | | Estimate differences[b] | 95% Cl | p-Value |
|---|---|---|---|---|---|
| Arms | How likely were you to engage patients in ACP **before** training? | How likely are you to engage patients in ACP **after** training? | | | |
| Team-based arm | 5.03 (0.26) | 7.95 (0.18) | 2.95 | 2.5; 3.42 | <0.001 |
| Individual clinician-focused arm | 6.11 (0.31) | 8.71 (0.21) | 2.60 | 2.06; 3.13 | <0.001 |
| Arms comparison[a] | | | 0.35 | -0.35; 1.07 | 0.31 |
| | Mean (SE) | | Estimate differences[b] | 95% Cl | p-Value |
| How likely are you to use the Serious Illness Conversations Guide (SICG) in your practice? | Part A[c] | Part B[d] | | | |
| Team-based arm | 8.05 (0.26) | 8.45 (0.15) | 0.46 | 0.12; 0.80 | 0.01 |
| Individual clinician-focused arm | 8.19 (0.29) | 8.68 (0.17) | 0.50 | 0.10; 0.90 | 0.02 |
| Arms comparison[a] | | | -0.04 | -0.57; 0.48 | 0.88 |

Analyzed using a linear mixed model.

p-value < 0.05 – statistically significant (t-test).

95% CI, confidence interval at 95%; SE, standard error of the mean.

[a]Arms comparison: [In team-based arm (Mean Part B-Mean Part A)] – [In Individual clinician-focused (Mean Part B- Mean part A)] or [In team-based arm (Mean After-Mean Before)] – [In Individual clinician-focused (Mean After-Mean Before)].

[b]Least squares mean.

[c]Part A was the online module.

[d]Part B was the role-play session

significant (p = 0.31). Likewise, mean difference between PHCPs' self-reported likelihood to use the SICG in their practice after Part A and after Part B was 0.46 (p = 0.01) in the team-based arm and 0.50 (p = 0.02) in the individual clinician-focused arm, the difference between arms was also not statistically significant (p = 0.88) (Table 5).

## Qualitative results

Fourteen facilitators were identified across seven of the 14 TDF domains, while 15 barriers were mapped onto nine of the TDF domains. The predominant barriers and facilitators were consistently associated with the *environmental contexts and resources* domain, with five barriers and six facilitators falling under this category. The second most prevalent facilitator was linked to the *skills* domain, while the second most common barrier was attributed to *social influences* domain. The most common theme among the barriers and facilitators in both arms was having enough time to have serious illness conversations. Having an organized workflow was an important facilitator in the team-based arm, but not in the individual clinician-focused arm. Similarly, communication problems with patients and PCHPs' own discomfort with serious illness conversations were more frequent barriers in the team-based arm than in the clinician-based arm. in addition, the team-based approach seems to have added barriers related to interprofessional coordination and interprofessional communication that were not an issue in the individual clinician-focused arm (Table 6).

## Triangulation

The psychosocial factors associated with CPD-Reaction matched the TDF domains of beliefs about capabilities, beliefs about consequences and social influences. Within the TDF, we identified eight additional psychosocial variables: knowledge, skills, reinforcements, behavioral

Table 6. *Mapping facilitators and barriers to the Theoretical Domains Framework (TDF) with illustrative quotes and frequencies by arms.*

| Facilitators, illustrative quotes and frequencies | | | |
|---|---|---|---|
| **TDF Domain** | What would help you have serious illness conversations and incorporate what you learned in this training in your practice? | Team-based arm (n = 223) | Individual clinician-focused arm (n = 150) |
| **Knowledge** | PHCPs knowing their patients and more able to identify those who needs serious illness conversations. "More information on their illness, prior to discussion" | 8 | 5 |
| **Skills** | More practice, including standardized patients "More practice…" | 33 | 30 |
| | More information, training, and practice scenarios (e.g., reference guide, videos, etc.) "More knowledge and training" | 9 | 7 |
| | Guidance on how to document serious illness conversations. "Sstructured documentation" | 6 | 2 |
| **Reinforcements** | Being reminded to have serious illness conversations. "Reminder sheets" | 7 | 10 |
| | Ongoing support "Follow-up of this training with the trainers in a few months" | 3 | 2 |
| | Billing "… billing…" | 2 | 0 |
| **Social Influences** | All team members understand the relevance of serious illness conversations and help each other. "…support and cooperation with other team members" | 28 | 0 |
| | Patients who are receptive and prepared to have serious illness conversations. "Patient responsiveness" | 5 | 3 |
| **Behavioral Regulation** | Holding multiple conversations "Breaking the conversation into multiple sessions makes so much sense" | 4 | 0 |
| **Environmental context and resources** | Having the Serious Illness Conversations Guide (SICG). "The guide is extremely helpful." | 23 | 31 |
| | Scheduled time designated for serious illness conversations. "Working toward scheduling dedicated visits to have serious illness conversations." | 21 | 12 |
| | Brief cheatsheet/buzzwords (instead of/in addition to the SICG) "A cheatsheet of phrases – buzzwords instead of a conversation guide" | 11 | 4 |
| | Having a clear workflow "Workflow to see how we can fit this into our primary care practice." | 26 | 4 |
| | Guides and templates embedded into electronic health records (EHRs) "Setting up templates in our EHR" | 8 | 5 |
| | More time to have these discussions. "Longer appointment times" | 37 | 26 |
| **Beliefs about capabilities** | Develop more confidence and experience to have serious illness conversations. "Having some help and gaining more experience in the field so I can feel more comfortable and confident" | 7 | 4 |
| **Others** | Unable to code[a] | 11 | 10 |
| | No answer | 44 | 31 |
| Barriers, illustrative quotes and frequencies | | | |
| | What barriers do you anticipate to incorporating what you learned in this training into your practice? | Team-based arm (n = 223) | Individual clinician-focused arm (n = 150) |

*(Continued)*

**Table 6.** (Continued)

| | Facilitators, illustrative quotes and frequencies | | |
|---|---|---|---|
| **Knowledge** | Not knowing the patient well and the difficulty in identifying the patients who would benefit. "Patients often don't see their primary care physician; we work with many patients we don't have a relationship with." | 11 | 10 |
| **Beliefs about consequences** | Patient's response "Patient being upset..." | 13 | 6 |
| **Emotion** | Discomfort with ACP and its emotional burden. "Having the conversation is out of my comfort zone." | 14 | 6 |
| **Social/ professional role and identity** | Scope of practice "I don't have as much direct patient care opportunity in my practice." | 7 | 2 |
| **Skills** | Lack of experience and practice "The inexperience" | 13 | 7 |
| **Environmental context and resources** | Workflow adjustments "Changing workflow to incorporate these interventions into practice" | 18 | 3 |
| | Documenting conversations "Once the conversation is done, what do we do with the answers." | 7 | 2 |
| | Unspecified barriers related to time "Time…" | 103 | 68 |
| | Fitting ACP into clinical schedule "Busy schedule. Making time for it because it is important." | 13 | 23 |
| | Difficulty in adapting the intervention to clinical routine. "Following script and checking all the boxes." | 10 | 16 |
| **Social Influences** | Involving and coordinating multiple personnel in the process "Lack of team concept at our practice." | 18 | 0 |
| | Lack of interprofessional communication about SICP "Communication with team" | 9 | 0 |
| | Communication issues with patients "Patients willing to have the conversation" | 23 | 14 |
| **Reinforcements** | Billing "… payment…" | 3 | 1 |
| **Goals** | Reaching out to patients "Starting to approach patients to invite them to have this conversation." | 6 | 0 |
| **Others** | Unable to code [a] | 8 | 3 |
| | No answer | 35 | 26 |

[a]We grouped under the 'Unable to Code' category all quotes where we could not understand what was written or which were not clear enough to be coded (e.g., "not sure", "none", etc.)

regulation, environmental context and resources, emotion, social/professional role and identity, and goals. The most frequent barrier and facilitator in both arms (i.e., environmental context and resources) was classified in the "opportunity" component of COM-B. Concurrently, comparative analysis of the arms using the COM-B revealed a large number of the barriers for the team-based arm derived from the domains of *opportunity* (social influences) and *motivation* (emotion and social/professional role and identity). Triangulating the results led to comprehensive recommendations (S2 Table). These were grounded in behavior change techniques linked to functions such as education, modelling, enablement, environmental restructuring, training, and persuasion. Notably, environmental restructuring (n = 7) and education (n = 6) emerged as the most frequently recommended functions (S2 Table).

## Discussion

This study is a mixed-methods theory-driven process evaluation embedded within a cRT that compared the impact on patients of a team-based SICP training approach for PHCPs with the impact of an individual clinician-focused SICP training approach, with post-intervention measures only. This process evaluation explored the causal mechanisms in operation relating to the PHCPs taking the training. Mechanisms that may have influenced the success or not of the intervention, and specifically PHCPs' intention to have serious illness conversations with patients. Contrary to the main hypothesis, intention was lower in the team-based (intervention) arm than in the individual clinician-focused arm, but after adjusting for age, profession and education this difference was not statistically significant. Concurrently, the self-reported likelihood to engage in ACP before training (assessed retrospectively) was lower in the team-based arm than in the individual clinician-focused arm. Both arms assessed likelihood to engage patients in ACP and to use the SICG as higher after training than before, but the mean difference between the likelihood before and after the training was no higher in the team-based arm than in the individual clinician-focused arm. Environmental context and resources were the most common TDF domain that emerged from the barrier and facilitators questions. Lack of time was a major barrier in both arms, while organized workflow was a key potential facilitator in the team-based arm. Additionally, the team-based arm reported more emotional discomfort and concerns about patients' reactions to serious illness conversations, as well as challenges coordinating multiple personnel and interprofessional communication that mapped onto the *social influence* TDF domain. The latter correlated with the lower mean on the team-based arm in the social influences CPD-Reaction construct. Triangulation of results using the COM-B framework showed most barriers and potential facilitators for both arms related to the *opportunity* domain. The COM-B model informed theory-based recommendations for improving future interprofessional SICP interventions. The recommendations primarily focused on environmental restructuring. These results lead us to the make the following observations:

First, contrary to the main hypothesis, a lower level of intention to have serious illness conversations with patients was found in the team-based arm than in the individual clinician-focused arm. However, the difference was no longer statistically significant when adjusted for age, profession and education. To the best of our knowledge, this is the first study to measure PHCPs' intention to have serious illness conversations with patients using the CPD-Reaction, so results cannot be compared with those of other studies. Yet, related evidence on ACP programs tends to favor team-based approaches [13,53]. The differences in the professions found in each arm of the parent cRT may explain these findings. This process evaluation also suggested that certain participants in the team-based arm may not, in their practices or healthcare systems, have had the deontological responsibility or professional expectations to have this type of discussion. Qualitative data too suggested that more PHCPs in the team-based arm perceived serious illness conversations as outside their usual scope of practice. Practices vary between countries, states, and provinces regarding integrating palliative care practices into primary care, and especially regarding who should initiate serious illness conversations [11,54,55]. For some of the other primary care team members this may have been the first time they considered engaging in such conversations [56]. In a survey of interprofessional healthcare professionals in Colorado, most agreed that it was exclusively the physician's role to discuss ACP with patients and family [57]. Future interprofessional trainings should take a more comprehensive approach, fostering support from managers or team leaders and using the expertise of more experienced PHCPs to encourage a sense of responsibility and confidence for holding serious illness conversations among those less experienced [54,55].

Second, PHCPs in both arms assessed their likelihood to engage in ACP before training as low, and the team-based arm rated it even lower than the individual clinician-focused arm. This could indicate that at study outset, baseline intention was low in the team-based arm, supporting the assumption that the lack of experience of certain team members in the team-based arm may have affected results. However, in both arms the training appeared to increase the likelihood of engaging in ACP and using the SICG, suggesting an overall positive impact of the SICP training sessions using either approach. A 2019 literature review on ACP in multiple settings also showed that training improved healthcare professionals' comfort with discussing end-of-life decisions [58]. Similarly, equipping PHCPs with the tools to have end-of-life discussions, such as the SICG, proved to be positive, as shown by the likelihood to use the SICG. However, the qualitative data showed there is interest in other tools too, such as cheatsheets and EHR-embedded tools. These findings are corroborated by research in other settings, where tools supplementing the SICG (e.g., SIC reminders, behavioral nudges) emerged as crucial elements in facilitating implementation [59–62]. Future adaptations of the SICP to primary care should prioritize equipping PHCPs with a wider range of tools which can be easily embedded in their clinical practice routines.

Third, the most frequent barriers and facilitators raised by PHCPs were related to the environmental context. For example, time emerged as both a barrier and a facilitator in both arms. We expected that a team-based approach would reduce the time PHCPs need for discussions by promoting the sharing of tasks. It is possible that team-based SICP still lacks better methods of sharing tasks to be more effective. A study that adapted the SICP to other contexts indicated that a well-organized division of tasks and workflow is an essential element for implementation of team-based approaches [13,61]. In this study participants indicated that team-based training needed to help teams organize their workflow and clarify and recognize the role of each professional. The lack of effect observed in the team-based approach could indeed be attributed to the additional time and burden required for coordination and communication among team members, as seen in qualitative data. Adaptations of the team-based SICP should prioritize designating tasks across roles and building profession-specific skills. In addition, participants in the team-based arm expressed more discomfort about serious illness conversations and had more doubts about patients' readiness for such discussions than participants in the individual clinician-focused arm. This finding supports our observation that despite the team-based training, there was a gap in certain participants' understanding and preparedness for engaging in serious illness conversations. Further efforts are needed to address their discomfort, fears of negative reactions and concerns about patients' preparedness.

Fourth, the CPD-reaction construct "social influences" exhibited the lowest mean score among the psychosocial factors influencing intention. This finding aligned with qualitative results from the team-based arm, where barriers such as interprofessional team coordination and communication difficulties were frequently mentioned and pertain to the domain of social influences within the TDF. "Social influence" is defined as the perception of approval or disapproval from significant peers or social referents regarding the adoption of a specific behavior [17]. One might expect PHCPs in the team-based training to report higher levels of social influence than those in the individual training group, but this was not the case. However, this finding aligns with other studies on the relationship between social influence and behavioral intention. A systematic review of 52 studies revealed that social influence had the lowest average score range among the four psychosocial factors influencing intention [63]. Nevertheless, Michie et al. established the social influence domain as a crucial target for health intervention planning [45], further supported by a systematic review analyzing 277 articles about behavior change techniques [64]. Improving social support measures in the

intervention therefore remains essential, including providing more information about peer approval and facilitating understanding and trust through social comparison [14,15].

Lastly, data triangulation revealed that the majority of barriers and facilitators to behavior adoption, especially in the team-based arm, stemmed from the *opportunity* domain of the COM-B framework, i.e., external physical or social factors in the environment that affect adoption of the behavior. The framework itself posits that opportunity influences capacity (encompassing intention), which in turn influences behavior (S1 Fig) [14,49]. Therefore, there remain upstream barriers and potential facilitators in the *Environmental context and resources* domain, that limit behavior change. Consequently, in line with the COM-B framework, modifications targeting environmental changes could improve the success of the intervention [32]. Training alone cannot address these issues. Structural adjustments within the clinical setting are crucial for facilitating such conversations. Incorporating these changes in team-based SICP training could help. However, this study encompassed primary care settings in two countries whose healthcare systems vary greatly, making it very difficult, for example, to integrate a "one-size-fits-all" workflow into the training or incorporate an EHR reminder into all practices. This finding highlights a critical challenge in large-scale interventions: balancing the need to standardize content across diverse clinical contexts with the need to adapt to each practice's specific needs [65–67]. Our findings, therefore, support the significance of environmental context modifications as effective tools for behavior change in future SICP implementations, wheter in research settings or healthcare policies.

## Limitations

A true pre-intervention measure of baseline intention to engage in serious illness conversations with patients would have been preferable, but pragmatic considerations precluded this. It is difficult to show differences in comparative effectiveness studies in real-world clinical practice, particularly when interventions are similar [68]. Second, this study is a process evaluation embedded within a cRT in which randomization units were the clinical practices and not the PHCPs. Although characteristics of the clinical practices were balanced between arms, PHCPs' characteristics were not balanced, as involving different professions in each group was part of the design. Therefore, meaningful comparison between arms for the main outcome, intention, could be limited. However, when controlled for these variables in the multivariable analysis the finding that intention increased more in the individual clinician-focused arm was no longer significant. This study analyzes team and clinician outcomes of a cRT for which the sample size and statistical power were calculated for patient outcomes, as these were the primary outcomes. Participation of PHCPs was encouraged, but the main focus was on patient recruitment and the number of practices, not the number of PHCPs. Thus, our results should be interpreted with caution, as their lack of significance may be secondary to a lack of statistical power. Future studies evaluating a team-based approach should therefore increase the number of team participants. Finally, the absence of semi-structured interviews or focus groups may have restricted the depth and complexity of qualitative data. This restriction stems from the time constraints imposed by real-world implementation projects. Despite this limitation, the open-ended question findings correlated with quantitative data and supported the study's conclusions.

## Conclusion

This theory-based process evaluation embedded within a cRT found that a team-based SICP training approach had no more impact than an individual clinician-focused training approach on increasing PHCPs' intention to have serious illness conversations with patients. However,

both training programs were useful for increasing PHCPs' likelihood of engaging in ACP discussions and using the SICG guide. A team-based approach requires further focus on training all team members to feel empowered to have serious illness conversations and should address their fears of negative reactions and their concerns about patients' preparedness. Our findings suggest that adaptations of the team-based SICP training approach should prioritize building more relevant skills and designating tasks across roles. Finally, our data triangulation suggested that in addition to training, structural environmental adjustments within clinical settings are crucial for facilitating serious illness conversations in teams in primary care. These findings will contribute to future interventions that improve the effectiveness of SICP training and support PHCPs in having more meaningful, high-quality and timely serious illness conversations with their patients.

## Supporting Information

**S1 Fig. The COM-B model and its correlation with the TDF domains.**
(DOCX)

**S1 Table. ACP engagement scores before and after training.**
(DOCX)

**S2 Table. Recommendations for improving the training based on barriers and facilitators, using the COM-B model, the Theoretical Domains Framework and the CPD-Reaction questionnaire.**
(DOCX)

**S1 Checklist. CONSORT**
(DOCX)

**S2 Checklist. GRAMMS.**
(DOCX)

## Acknowledgments

We acknowledge the precious work of Louisa Blair for her editorial help with the manuscript and Stéphane Turcotte for his vital insights in the statistical analysis. We also thank the members of the Meta-LARC ACP cRT team for their involvement in this project.

COLLABORATORS META-LARC ACP TRIAL TEAM:

Angela K. Combe, Oregon Health & Science University

Annette M. Totten, Oregon Health & Science University

Barcey T. Levy, University of Iowa

Cat Halliwell, University of Colorado

David A. Dorr, Oregon Health & Science University

David Nowels, University of Colorado

Deb Constien, Patient partner

Deborah Dokken, Patient partner

Donald E. Nease, Jr., University of Colorado

Dr. B. Angeloe Burch Sr., Patient partner

Elizabeth Fernley, Oregon Health & Science University

France Légaré, Université Laval and VITAM - Centre de recherche en santé durable

Gail Drey, Patient partner

Gurnoor Kaur Brar, University of Toronto

Jacqueline D. Alikhaani, Patient partner

James Pantelas, Patient partner

Jean-Sébastien Paquette, Université Laval and VITAM - Centre de recherche en santé durable

Jeanette M. Daly, University of Iowa

Jessica E. Ma, Duke University

Jodi Lapidus, Oregon Health & Science University - Portland State University School of Public Health

Judy Katz, Patient partner

Kate Hanrahan, University of Iowa

Kathy Kastner, Patient partner

Katrina Ramsey, Oregon Health & Science University

Keith Provin, Patient partner

Kirsten Wentlandt, University Health Network

Kylie Lanman, Oregon Health & Science University

LeAnn C Michaels, Oregon Health & Science University

Lyle J. Fagnan, Oregon Health & Science University

Mary F. Henningfield, PhD, University of Wisconsin-Madison

Mary M. Minniti, Patient partner

Matthew Howard, Oregon Health & Science University

Megan Schmidt, University of Iowa

Meredith K. Warman, University of Colorado

Michelle Greiver, University of Toronto

Olga Petrova, Patient partner

Patrick M. Archambault, Université Laval, VITAM - Centre de recherche en santé durable, and Centre de recherche intégrée pour un système apprenant en santé et services sociaux de Chaudière-Appalaches

Peter Kim, University of Iowa

Rowena J. Dolor, Duke University

Sabrina Guay-Bélanger, VITAM - Centre de recherche en santé durable

Sarah Bumatay, Oregon Health & Science University

Sarina Schrager, University of Wisconsin-Madison

Sean Rice, Oregon Health & Science University-Portland State University School of Public Health

Sharon E. Straus, University of Toronto

Shelbey Hagen

Shigeko (Seiko) Izumi, Oregon Health & Science University

Souleymane Gadio, VITAM - Centre de recherche en santé durable

Suélène Georgina Dofara, VITAM - Centre de recherche en santé durable

Susan Lowe, Oregon Community Health Information Network - Columbia Gorge Health Council

Taryn Bogdewiecz, University of Colorado

## Author contributions

**Conceptualization:** Sabrina Guay-Bélanger, LeAnn Michaels, Jean-Sébastien Paquette, Shigeko (Seiko) Izumi, Annette M. Totten, France Légaré.

**Data curation:** Lucas Gomes Souza, Patrick M. Archambault, Suélène Georgina Dofara, LeAnn Michaels, Jean-Sébastien Paquette, Shigeko (Seiko) Izumi, Annette M. Totten, France Légaré.

**Formal analysis:** Lucas Gomes Souza, Patrick M. Archambault, Dalil Asmaou Bouba, Suélène Georgina Dofara, Sergio Cortez Ghio, Souleymane Gadio, Shigeko (Seiko) Izumi, Annette M. Totten, France Légaré.

**Funding acquisition:** Sabrina Guay-Bélanger, LeAnn Michaels, Annette M. Totten, France Légaré.

**Investigation:** Lucas Gomes Souza, Patrick M. Archambault, Suélène Georgina Dofara, France Légaré.

**Methodology:** Lucas Gomes Souza, Patrick M. Archambault, Dalil Asmaou Bouba, Suélène Georgina Dofara, Sabrina Guay-Bélanger, Shigeko (Seiko) Izumi, Annette M. Totten, France Légaré.

**Project administration:** Suélène Georgina Dofara, Sabrina Guay-Bélanger, LeAnn Michaels, Annette M. Totten, France Légaré.

**Resources:** Sabrina Guay-Bélanger, France Légaré.

**Software:** Sergio Cortez Ghio, Souleymane Gadio, France Légaré.

**Supervision:** Patrick M. Archambault, Suélène Georgina Dofara, Sabrina Guay-Bélanger, LeAnn Michaels, Annette M. Totten, France Légaré.

**Validation:** Lucas Gomes Souza, Patrick M. Archambault, Suélène Georgina Dofara, Annette M. Totten, France Légaré.

**Visualization:** Patrick M. Archambault, Suélène Georgina Dofara, France Légaré.

**Writing – original draft:** Lucas Gomes Souza, France Légaré.

**Writing – review & editing:** Lucas Gomes Souza, Patrick M. Archambault, Dalil Asmaou Bouba, Suélène Georgina Dofara, Sabrina Guay-Bélanger, Sergio Cortez Ghio, Souleymane Gadio, LeAnn Michaels, Jean-Sébastien Paquette, Shigeko (Seiko) Izumi, Annette M. Totten, France Légaré.

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
