## [Decision Letter · Decision Letter 0]

30 Sep 2024

PONE-D-24-03149Impact of a team-based versus individual clinician-focused training approach on primary healthcare professionals’ intention to have serious illness conversations with patients: a theory informed process evaluation of a cluster randomized trial.PLOS ONE

Dear Dr. Légaré,

Thank you for submitting your manuscript to PLOS ONE. After careful consideration, we feel that it has merit but does not fully meet PLOS ONE’s publication criteria as it currently stands. Therefore, we invite you to submit a revised version of the manuscript that addresses the points raised during the review process.

We look forward to receiving your revised manuscript.

Kind regards,

Mohammed Hussain Abutaleb, PhD

Academic Editor

PLOS ONE

**Journal Requirements:**

2. We note that your article has been submitted as a "Clinical Trial" article type, but it is a Research Article. When resubmitting your manuscript, we ask that you update your article type to "Research Article" in the online submission form. Please note that some fields in the submission form may be reset with this change, so please go through your submission in full to ensure that all information is accurate and complete when resubmitting your manuscript.

Reviewers' comments:

Reviewer's Responses to Questions

**Comments to the Author**

1. Is the manuscript technically sound, and do the data support the conclusions?

Reviewer #1: Yes

Reviewer #2: Partly

2. Has the statistical analysis been performed appropriately and rigorously? 

Reviewer #1: Yes

Reviewer #2: No

3. Have the authors made all data underlying the findings in their manuscript fully available?

Reviewer #1: Yes

Reviewer #2: Yes

4. Is the manuscript presented in an intelligible fashion and written in standard English?

Reviewer #1: Yes

Reviewer #2: No

5. Review Comments to the Author

**Reviewer #1: ** Although the the manuscript has the intention to evaluate the cRTs but the real outcome as per title is the intention serious illness conversations with patients (SICP) was not addressed in the background of abstract. There is need to address the gaps and magnitude of the intention SICP as the problem statement in the introduction and more issues to be highlighted on the outcome. Methodology was well elaborated and the results were well discussed.

**Reviewer #2: ** There is a need to shorten the article.Use academic language throughout the document.

Refrain from usage of ‘we’ (as in abstract), repeated several times.

Figure S1 needs to be improved on or removed.

The Meta-Larc Appendix has the term advanced care planning, whereas the title here is serious illness.

There is a need to improve on the clarity of the whole manuscript. It appears rather difficult to read through.

6. PLOS authors have the option to publish the peer review history of their article (what does this mean? ). If published, this will include your full peer review and any attached files.

**Do you want your identity to be public for this peer review?** For information about this choice, including consent withdrawal, please see our Privacy Policy .

Reviewer #1: No

Reviewer #2: No

---

## [Author Response · Author response to Decision Letter 0]

14 Nov 2024

Dear reviewers,

Thank you for your valuable feedback on our paper. Your comments are greatly appreciated, and we have carefully considered each point raised. Below, you will find our detailed responses to address your suggestions:

Reviewer 1

Comment: Although the manuscript has the intention to evaluate the cRTs, the real outcome as per the title—the intention for serious illness conversations with patients (SICP)—was not addressed in the background of the abstract. There is a need to address the gaps and the magnitude of the intention for SICP as the problem statement in the introduction, and more issues need to be highlighted on the outcome.

Response: Dear reviewer, thank you for this comment, which has helped us clarify in the paper how measuring PHCP intention functioned as a process evaluation. We have rewritten the background of the Abstract as follows:

“Cluster randomized trials (cRTs) on the effectiveness of training programs face complex challenges when conducted in real-world settings. Process evaluations embedded within cRTs can help explain their results by exploring possible causal mechanisms impacting training effectiveness.”

We also added this paragraph in the Introduction to better explain this:

Process evaluations embedded within cRTs aim to provide insights into how an intervention was delivered, how participants received it, and whether it was implemented as intended (French, Pinnock et al. 2020). In the parent Meta-LARC ACP cRT, the impact of a team-based training approach compared to a clinician-based training approach was measured using patient-centered outcomes (Totten, Fagnan et al. 2019). In this process evaluation, measuring the impact of the intervention on PHCPs themselves instead of patients and, specifically, on PHCPs’ intention to have serious illness conversations with patients was necessary to provide insights into how the team-based training approach was delivered, received, and implemented in the parent cRT. A process evaluation focusing on PHCPs’ behavioral intentions and its psychosocial determinants would thus provide information about key factors affecting the effective implementation of the intervention and could inform recommendations for its improvement. Key factors would include modifiable psychosocial factors that influence PHCPs’ intention to have serious illness conversations, as well as the barriers and facilitators they perceive to implementing these conversations. Thus, to influence behavior through behavioral intention, one would need to rely on modifying psychosocial variables using known behavior change techniques (Michie, Atkins et al. 2014).

We also added the following paragraph to highlight the knowledge gap that motivated our study and detail the magnitude of intention:

No studies have specifically addressed the factors that drive PHCPs’ intentions to have serious illness conversations with patients. Understanding these components is crucial for designing training interventions that effectively promote the adoption of this important behavior in clinical practice. According to several socio-cognitive theories, behavioral intention is the central factor influencing the adoption of a given behavior and is therefore an acceptable surrogate for behavior change (Godin, Bélanger-Gravel et al. 2008, Hrisos, Eccles et al. 2009, Godin 2013, Légaré, Borduas et al. 2014, Bakwa Kanyinga, Gogovor et al. 2023). It provides a more immediate and cost-effective measure of actual clinical behavior change and is widely used to evaluate the efficacy of professional development training interventions (Goulet, Hudon et al. 2013). Measuring intention would thus allow for a targeted assessment of the intervention's influence on PHCPs, the primary implementers of the intervention in the parent cRT. It could also provide information on the intervention’s strengths and weaknesses to inform the interpretation and adaptation of future SICP interventions (Grimshaw, Presseau et al. 2014). Ultimately, it could contribute to a cumulative implementation science knowledge base.

Page: Pages 3 and 6

Comment: Methodology was well elaborated, and the results were well discussed.

Response: We appreciate your feedback.

Reviewer 2

Comment: There is a need to shorten the manuscript. It appears rather difficult to read through.

Response: Dear reviewer, thank you for your valuable feedback. We have shortened the manuscript considerably (approximately 800 words) and edited it for improved readability. We hope these changes make it easier to follow and understand.

Comment: Use academic language throughout the document. Refrain from usage of ‘we’ (as in abstract), repeated several times.

Response: Thank you for your comment. We appreciate your suggestions for improving the academic tone of our manuscript. We've carefully reviewed your comments and revised the document to eliminate all first-person pronouns and avoid any informal language.

Comment: Figure S1 needs to be improved on or removed.

Response: We appreciate your feedback regarding Figure S1. After careful consideration, we have decided to remove the figure. We believe that the information presented in the figure is sufficiently covered in the preceding paragraph, which cites references 17 and 18 to elaborate on how behaviors are embedded in a complex system. Additionally, we have provided a detailed explanation of how the variables were collected, ensuring that the necessary context is conveyed without the need for the supplemental figure.

Page: Page 10

Comment: The Meta-Larc Appendix has the term “advanced care planning,” whereas the title here is “serious illness.”

Response: Dear reviewer, thank you for this comment. We agree that the use of the two terms can be confusing. The title "Meta-LARC ACP" is the name consistently used to refer to the parent trial. This term was adopted in various scientific communications and is the trial's official name in current usage:

https://doi.org/10.1016/j.jpainsymman.2022.04.019

https://doi.org/10.1016/j.jpainsymman.2023.02.297

Doi: 10.35680/2372-0247.1808.

There is, however, an ongoing debate in palliative care concerning the terminology—whether "serious illness conversations" (SIC) or "advance care planning" (ACP) is more appropriate (doi:10.1001/jama.2021.23695 or 10.1111/jgs.16801). The original trial that developed the Serious Illness Care Program framed serious illness conversations as a tool for ACP (10.1136/bmjopen-2015-009032).

However, to provide a more comprehensive understanding of behaviors related to end-of-life care, our research incorporated both ACP and SIC as complementary concepts and behaviors. This approach avoids potential controversies and allows for a deeper analysis of the theme. As Lou, Atkins, and West suggest (https://doi.org/10.1186/s13012-017-0605-9), behaviors are part of a broader system and cannot be isolated. By progressively broadening the scope of behaviors assessed—in this case, using both SIC and ACP—we succeeded in enhancing both the comprehensiveness and detail of our analysis. We believe this explanation clarifies our rationale for combining ACP and SIC in our research.

However, to further clarify this difference for our readers, we have added the following sentence in the Outcomes and Measurements section (Methods):

“ACP was included because it is a more established, broader concept in the field that encompasses end-of-life discussions, care objectives, and their legal documentation (Sudore, Hickman et al. 2022).”

Comment: There is a need to improve on the clarity of the whole manuscript.

Response: Thank you for your valuable feedback. We’ve addressed your suggestion by having a professional scientific editor review the manuscript, enhancing its clarity, conciseness, and academic tone. We believe these revisions have greatly improved readability and facilitated reader comprehension.

---

## [Decision Letter · Decision Letter 1]

15 Jan 2025

Impact of a team-based versus individual clinician-focused training on primary healthcare professionals’ intention to have serious illness conversations with patients:a theory-informed process evaluation embedded within a cluster randomized trial

PONE-D-24-03149R1

Dear Dr. Légaré,

We’re pleased to inform you that your manuscript has been judged scientifically suitable for publication and will be formally accepted for publication once it meets all outstanding technical requirements.

Kind regards,

Mohammed Abutaleb, PhD

Academic Editor

PLOS ONE

Additional Editor Comments (optional):

Reviewers' comments:

Reviewer's Responses to Questions

**Comments to the Author**

1. If the authors have adequately addressed your comments raised in a previous round of review and you feel that this manuscript is now acceptable for publication, you may indicate that here to bypass the “Comments to the Author” section, enter your conflict of interest statement in the “Confidential to Editor” section, and submit your "Accept" recommendation.

Reviewer #1: All comments have been addressed

Reviewer #3: All comments have been addressed

2. Is the manuscript technically sound, and do the data support the conclusions?

Reviewer #1: Yes

Reviewer #3: Yes

3. Has the statistical analysis been performed appropriately and rigorously? 

Reviewer #1: Yes

Reviewer #3: Yes

4. Have the authors made all data underlying the findings in their manuscript fully available?

Reviewer #1: Yes

Reviewer #3: Yes

5. Is the manuscript presented in an intelligible fashion and written in standard English?

Reviewer #1: Yes

Reviewer #3: Yes

6. Review Comments to the Author

Reviewer #1: I believe that this paper is now ready for publishing because the authors have sufficiently addressed my concerns from a prior review round.

Reviewer #3: (No Response)

7. PLOS authors have the option to publish the peer review history of their article (what does this mean? ). If published, this will include your full peer review and any attached files.

**Do you want your identity to be public for this peer review?** For information about this choice, including consent withdrawal, please see our Privacy Policy .

Reviewer #1: No

Reviewer #3: **Yes: ** Esedullah AKARAS

---

## [Editor Report · Acceptance letter]

PONE-D-24-03149R1

PLOS ONE

Dear Dr. Légaré,

I'm pleased to inform you that your manuscript has been deemed suitable for publication in PLOS ONE. Congratulations! Your manuscript is now being handed over to our production team.

Kind regards,

on behalf of

Dr. Mohammed Abutaleb

Academic Editor

PLOS ONE